# Different carotenoid conformations have distinct functions in light-harvesting regulation in plants

Nicoletta Liguori[1], Pengqi Xu[1], Ivo H.M. van Stokkum [1], Bart van Oort [1], Yinghong Lu[2,3], Daniel Karcher[2], Ralph Bock[2] & Roberta Croce [1]

To avoid photodamage plants regulate the amount of excitation energy in the membrane at the level of the light-harvesting complexes (LHCs). It has been proposed that the energy absorbed in excess is dissipated via protein conformational changes of individual LHCs. However, the exact quenching mechanism remains unclear. Here we study the mechanism of quenching in LHCs that bind a single carotenoid species and are constitutively in a dissipative conformation. Via femtosecond spectroscopy we resolve a number of carotenoid dark states, demonstrating that the carotenoid is bound to the complex in different conformations. Some of those states act as excitation energy donors for the chlorophylls, whereas others act as quenchers. Via in silico analysis we show that structural changes of carotenoids are expected in the LHC protein domains exposed to the chloroplast lumen, where acidification triggers photoprotection in vivo. We propose that structural changes of LHCs control the conformation of the carotenoids, thus permitting access to different dark states responsible for either light harvesting or photoprotection.

[1] Department of Physics and Astronomy and Institute for Lasers, Life and Biophotonics, Faculty of Sciences, Vrije Universiteit Amsterdam, De Boelelaan 1081, 1081 HV Amsterdam, The Netherlands. [2] Max-Planck-Institut für Molekulare Pflanzenphysiologie Wissenschaftspark Golm, Am Mühlenberg 1, 14476 Potsdam-Golm, Germany. [3] Present address: School of Chemical Engineering, Nanjing University of Science and Technology, Nanjing, China. Correspondence and requests for materials should be addressed to R.C. (email: r.croce@vu.nl)

I n photosynthetic eukaryotes, the pigment–protein complexes responsible for harvesting sunlight and transferring excitation energy to the photosystems can dynamically modulate their efficiency in response to changes in sunlight intensity[1,2]. Regulating the level of excitation in the photosynthetic membrane is essential for the fitness of plants and algae because, under strong light intensities, excess chlorophyll (Chl) excitation energy may lead to photooxidative damage[3].

It has been shown that these antennae, known as light-harvesting complexes (LHCs), can exist in different conformations corresponding to distinct functional states: highly fluorescent states, essential for light-harvesting, or dissipative (quenched) states, potentially useful for photoprotection[4,5]. Therefore, it has been proposed that the fastest response of plants and algae to highlight stress consists in stabilizing LHCs in their quenched conformation(s)[2]. However, the possible structural change(s) involved in the conformational switch[5–7] and also the nature of the quenching mechanism itself[8–13] have not yet been elucidated.

The molecular design of LHCs is based on an integral membrane apoprotein (≈25 kDa) binding a high concentration of Chls and carotenoids (Cars)[14,15]. The protein matrix is responsible for maintaining and possibly rearranging distances and orientations of the pigments relative to each other. Indeed, as recently shown in silico and in vitro, distinct conformations of single protein domains correlate with different energy states in the antennae[5–7], suggesting that structural changes of the protein matrix are responsible for the switch to photoprotective states in the LHCs. Yet, it is unknown which changes in the chromophores, possibly coupled to rearrangement(s) in the protein, are necessary to create quenching site(s). Chls and Cars display a high degree of flexibility, in the form of distortions and/or displacements, whereas bound to LHCs[5,7,16,17]. Raman studies have shown that twisted conformations of the Cars bound to LHCs correlate with dissipative states of these complexes[5,18]. However, a direct role of Cars distortions in creating quenching sites within LHCs has not been established so far.

Although it is generally accepted that the origin of quenching within LHCs can be ascribed to specific interactions between the pigments bound to them, the exact nature of such interactions is still under debate[8–11,13]. A variety of potential quenching mechanisms has been proposed, each resulting in shortening of Chl singlet excited state ($^1$Chls*) lifetime via fast (≈ ps) thermal dissipation. Strongly coupled Chls, for example, have been shown to possess the ability to dissipate energy by forming charge-transfer states[10,19,20]. All other proposed quenching channels involve Chl–Car interactions and have been found to correlate with modulation of fluorescence within the thylakoid[11] and in isolated complexes[8,9,13]. Cars are indeed excellent candidates for dissipating $^1$Chls* excitation energy due to their ultra-short excited state lifetime (~10 ps) and to the presence of low-lying singlet energy levels in their energy landscape[21–23]. However, the role in photoprotection of these so-called dark states of Cars, as they are forbidden for one-photon transitions, is still under debate[8–10].

Here, we have used a model LHC which is constitutively present in an energy-dissipative state. This LHC retains the same protein and Chl structural organization of the native monomeric LHCs but contains only a single Car species. Thanks to this pigment composition, by combining ultrafast spectroscopy with a compartment model fitting, we can discriminate the functions that different conformations of the same pigment bound to the antenna can exert. We can, therefore, explore the possible structure-function relationship of the Cars involved in quenching in LHCs. By combining these results with molecular dynamics simulations we demonstrate that photosynthetic light harvesting is controlled by conformational changes in the Cars bound to the LHCs.

## Results

**LHC–Asta is strongly quenched.** Expression of the biochemical pathway for the synthesis of the ketocarotenoid astaxanthin (Asta) from the chloroplast genome of tobacco (*Nicotiana tabacum*) has been shown to result in high levels of astaxanthin[24]. We have further optimized the expression of the pathway enzymes by using a recently developed synthetic operon design[25]. This approach resulted in tobacco plants that channeled the entire flux through the Car pathway for astaxanthin production[26]. As a consequence, the LHCs of these plants do not contain the native Cars (lutein, violaxanthin, neoxanthin, and zeaxanthin)[27,28] but only the ketocarotenoid astaxanthin. At variance with wild-type plants, trimeric LHCs are not present in this mutant because the absence of lutein impairs the capacity of LHCII to form trimers[29]. As a result, the purified LHCs (subsequently referred to as LHC–Asta) are exclusively monomeric and contain mostly LHCII, which is the most abundant antenna of plants[14].

The Chl a/b ratio of 1.33 found in LHC–Asta is identical to that of wild-type LHCII (LHCII-WT), suggesting that the presence of astaxanthin does not affect Chl binding. In all monomeric LHCs[30,31], up to three Car-binding sites are occupied, namely the internal L1, L2, and the external N1 (see Fig. 1c). If we assume 14 Chls per monomeric LHC as in LHCII-WT[27], the Chl/Car ratio of 2.37 measured in LHCII-Asta indicates that at least three molecules of the total astaxanthin pool are functionally unconnected to the complex.

LHC–Asta and LHCII-WT are characterized by a highly similar absorption spectrum in the Chl–$Q_X$/$Q_Y$ region (Fig. 1a), with a minor difference in the Chl b peak possibly owing to the exchange neoxanthin/astaxanthin, which might influence the absorption of the Chl b cluster close to the N1 site[32]. In contrast, the Soret/Car region differs substantially due to difference in the Cars content. The strong Chl–Chl excitonic interactions of LHCII-WT are preserved in LHC–Asta, as testified by the similar amplitude and position of the peaks in the circular dichroism spectrum (Fig. 1b)[33]. Together, these data suggest that the overall Chl organization of LHC–Asta is very similar to that of LHCII-WT.

At variance with the wild-type, LHC–Asta is constitutively in an energy-dissipative state, as shown by the strong reduction (72%) of its fluorescence quantum yield relative to LHCII-WT (Fig. 1d, Supplementary Table 1), permitting the study of the quenching process at the level of single LHC complexes in a native-like condition.

**Excitation energy transfer and quenching in LHC–Asta.** We investigated the origin of the quenching in LHC–Asta by means of femtosecond transient absorption with selective excitation of either astaxanthin (at 508 ± 5 nm) or Chls (at 690 ± 5 nm) (see Supplementary Fig. 1). In this way, we probed excitation energy transfer (EET) from astaxanthin to Chls and vice versa.

A first impression of the EET dynamics was obtained by fitting the data with a sequential kinetic scheme (global analysis)[34]. In this simplified picture, every state decays exponentially into the following one. By applying this model, we retrieved spectra associated with each evolving species (evolution-associated difference spectra, EADS). From the numerical equivalent parallel decay model, we obtained the decay-associated difference spectra (DADS). From the EADS and the DADS we identified the rise and/or decay of the different excited states. Here, we first present the main observations obtained from the global analysis. A full description of the different components retrieved via global

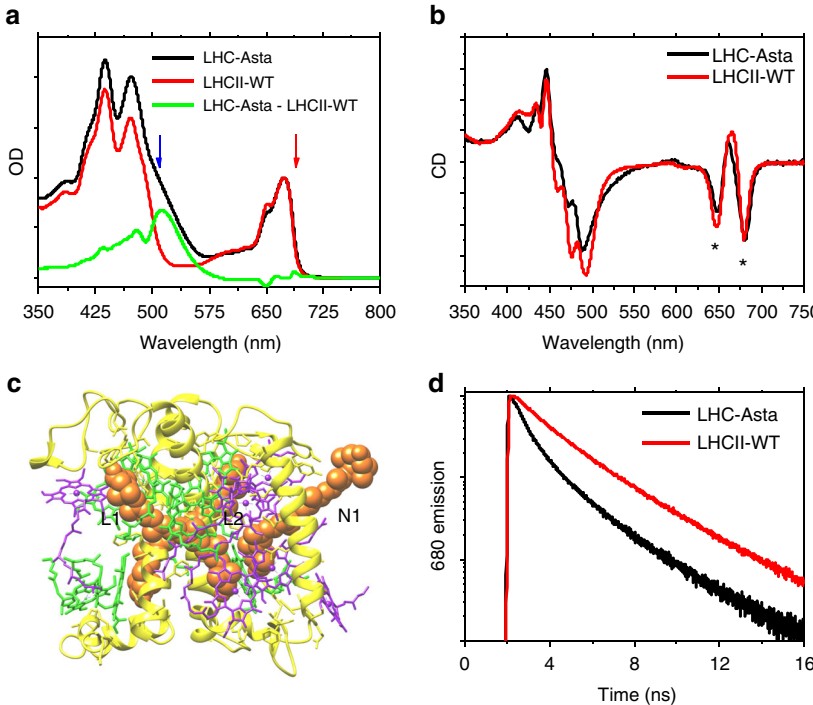

**Fig. 1** Spectral characterization of LHC–Asta and comparison with LHCII-WT. **a** Steady-state absorption spectra (normalized to the Chl content, assumed to be the same in LHC–Asta and LHCII-WT). The difference absorption spectrum (LHC–Asta minus LHCII-WT; green curve) shows features of astaxanthin in solution[35] minus the Cars missing in LHCII-Asta (lutein, neoxanthin, and violaxanthin)[67]. The blue and the red arrows indicates the two excitation wavelengths used for the transient absorption experiments (respectively, 508 and 690 nm). **b** Circular dichroism (normalized to the OD) of LHC–Asta and LHCII-WT monomers. The asterisks indicate the conserved Chl excitonic bands. **c** A model of the protein and pigment organization of this complex based on the estimated pigment stoichiometry (about 14 Chls and three Cars per monomer, see text). In the model the protein is depicted in yellow, the Chls a and b in green and purple, respectively, and the astaxanthin molecules in orange. Native Car-binding sites in monomeric LHCs are also indicated (L1, L2, N1). **d** Fluorescence decay kinetics (normalized to peak value) detected at 680 nm after excitation at 470 nm, y-axis in logarithmic scale

analysis is given in the caption of Supplementary Fig. 2a–d. The quantitative model (target analysis)[34] of the EET and quenching dynamics in LHC–Asta are presented in the next paragraph.

The evolution of the astaxanthin-excited states dynamics upon 508 nm excitation is described by at least 6 components (Fig. 2a and Supplementary Fig. 2a, b). The first prominent feature visible in this analysis concerns the presence of multiple dark states in the energy landscape of this Car. The spectrum appearing with a time constant of less than 200 fs (red EADS in Fig. 2a) has the characteristics of the spectrum that was assigned to the dark state S1 in a previous work on monomeric astaxanthin in dimethyl sulfoxide[35] (i.e., the ground state bleach at ~500 nm and the two distinct positive peaks of excited state absorption at ~595 and ~630 nm). However, although in isolated astaxanthin this state displays a mono-exponential decay (5.3 ps)[35], the decay in LHC–Asta is bi-exponential with 2.7 and 7.8 ps time constants. The 7.8 ps EADS (Fig. 2a, green) resembles the S1 state spectrum, but the relative amplitude of the two maxima (595 and 630 nm) has changed. These features suggest that the decay of astaxanthin takes place in the ps time range via two different low-energy states characterized by different spectra and lifetimes, with the longer one assigned to the species peaking at ~595 nm. The EADS also show that at least a sub-population of the astaxanthin bound to LHC–Asta is active in EET to Chls. This is visible as a rise of the Chl bleach around 680 nm on the timescale of astaxanthin decay (see also the two positive DADS around 680 nm, Supplementary Fig. 2a). The excitation spectrum of LHC–Asta (Supplementary Fig. 3) confirms that there is EET from astaxanthin to Chls. The different lifetimes at which the Chl signal appears are compatible with the decay of S2 (assigned based on the spectra and lifetimes[35]) (≪100 fs), vibrationally

"hot" S1 (170 fs) and S1 (2.7 ps) states of astaxanthin, suggesting that all those states are involved in EET to Chls. In contrast, the 7.8 ps decay of the species peaking at ~595 nm is not accompanied by any visible EET to the Chls.

The evolution of the Chl-a decay, after preferential excitation in the red edge of the $Q_Y$ region (690 nm) can be described by at least four components and the results are reported in Fig. 2b (normalized DADS) and Supplementary Fig. 2b (EADS). The initial EADS (Supplementary Fig. 2b) shows ground state bleach peaking at ≈680 nm (band associated to Chl a-$Q_Y$) and has an associated lifetime of 2.2 ps, which is typical for EET between Chls in LHCs (e.g. ref. [36]). Notably, on this timescale, ~20% of the energy is lost. The explanation for this energy loss is provided by the normalized DADS in Fig. 2b: in the case of EET exclusively between Chl species, all DADS would resemble pure Chl spectra. However, the black DADS associated with the 2.2 ps component does not resemble that of pure Chl excited states. It can be inferred that two phenomena take place simultaneously in the first 2.2 ps: (1) energy equilibration between the most red Chls (preferentially excited at 690 nm) and the blue ones (see inset in Fig. 2b and Supplementary Fig. 2b) and (2) EET to an additional species that is characterized by a spectrum different from that of Chls. Interestingly, in the 480–640 nm region, the difference between the 2.2 ps DADS and a pure Chl DADS resembles that of the inverted green, and blue EADS reported in Fig. 2a, which we assigned to astaxanthin singlet excited state(s).

To resolve the spectra of the Car and Chl pools of LHC–Asta active in EET and in quenching, we applied a compartmental target fit model[34]. Each compartment represents a species with its species-associated difference spectrum (SADS). With a small set of spectral assumptions, decay and transfer rates between

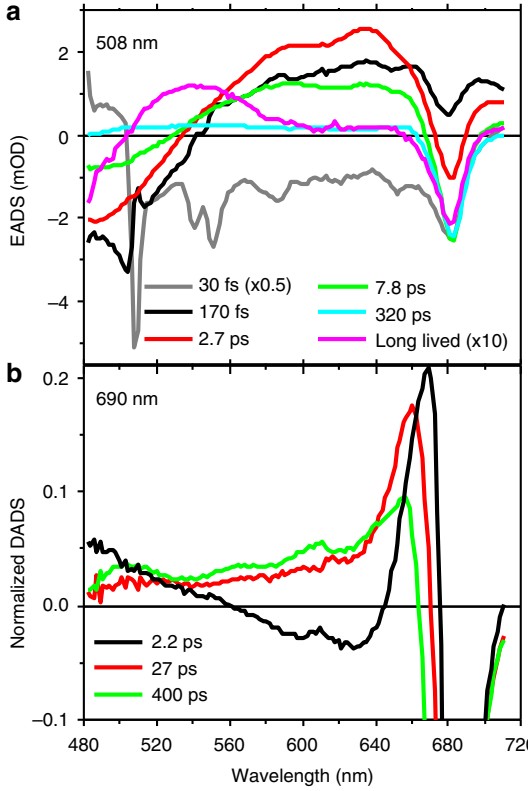

**Fig. 2** Results of the global analysis after selective excitation of Cars and Chls. **a** EADS estimated from the global analysis of the data collected upon 508 nm (Cars) excitation. The gray and magenta EADS associated to ultrafast (<100 fs) and long-lived (>ns) species, respectively, have been scaled for clarity as indicated in the legend. Given the short timescale (30 fs), coherent artifacts are likely present in the gray spectrum. **b** First three DADS normalized to their minimum for the data collected upon 690 nm (Chls) excitation. The full set of EADS and DADS for both excitation wavelengths is reported in Supplementary Figure 2a, b

compartments can be estimated. The quality of the fit of the simultaneous analysis of all data is excellent (see Supplementary Fig. 4). In the following, we first describe the model of the EET dynamics in LHC–Asta and then the quenching process.

The kinetic scheme used for the analysis of the dataset collected upon 508 nm excitation (Cars excitation) is shown in Fig. 3b and the estimated SADS are shown in Fig. 3a. The model also includes excited states dynamics of the astaxanthin molecules that are unconnected to Chls in LHC–Asta (omitted from Fig. 3b for clarity). The complete model is presented in Supplementary Figs 5 and 6 together with all a priori assumptions on which it is based. The concentrations of the different species estimated via target analysis are presented in Supplementary Fig. 7.

Because of the heterogeneity of lifetimes and spectra observed via global analysis (see above), the ultrafast decay of the S2 state was modeled to take place via two competing channels. The first channel is represented by "hot" S1 (green SADS in Fig. 3a), which then relaxes to S1 (blue SADS). The SADS of "hot" S1 presents the features of a vibrationally unrelaxed S1 with the ground state bleach centered at ~500 nm and the excited state absorption extended above 700 nm. This resembles the shape of a red-shifted S1 spectrum, similar to monomeric astaxanthin[35]. The second channel leads to a compartment with a spectrum (red SADS) that is different from S1 and hot S1 and therefore corresponds to a different electronic state. Based on the position of the maxima of the red SADS, which are at higher energy than that of S1, and on

its lifetime, which is longer than that of S1 (8 ps ($133 ns^{-1}$) vs. 6 ps ($183 ns^{-1}$)), we assign this species to the so-called S* state of astaxanthin. Indeed similar properties characterize the S* spectrum and lifetime of Cars in bacterial LHs[37,38]. Excited astaxanthin further depopulates via EET to Chls via S2, hot S1, and S1 with associated rate constants (7000, 3000, and $366 ns^{-1}$) comparable to the ones estimated for lutein to Chl in wild-type LHCII[39].

The model applied to the measurements collected upon Chl excitation (690 nm) includes six compartments (Fig. 3c, d; full model in Supplementary Fig. 5b).

Equilibration between Chls occurs at multiple timescales, from the lowest energy Chl (Chl a1', blue SADS in Fig. 3c) via an intermediate (Chl a2', gold) to the highest energy Chl (Chl a3', orange SADS). Equilibration is "uphill", because the 690 nm laser preferentially excites low-energy Chls. The SADS of Chl a1', a2' and a3' show very similar excited state absorption, which is nearly flat in the region 480–620 nm, without imposing any spectral constraint.

The quenching process is modeled for both excitations as a dissipative pathway in which all the Chls populate an unknown species called $S_q$ (Fig. 3b, d). The majority of Chls is quenched by $S_q$ on multiple timescales, with estimated rates ranging between 1.59 and $5.07 ns^{-1}$. Only a minor part of the excited Chl pool leads to the formation of Chl and Car triplets (see Supplementary Fig. 5a, b). It is worth noting that the spectrum and the lifetime of $S_q$ were estimated independently from all other species. The spectrum of $S_q$ (black SADS in Fig. 3c) displays a broad excited state absorption between ~540 and ~650 nm similarly to the dark states of astaxanthin, S1 and S* (blue and red spectra in Fig. 3a). The lifetime of $S_q$ is estimated to be ~12 ps ($84 ns^{-1}$), comparable again to that of a Car dark state.

**The nature of the quencher in LHC–Asta.** To better elucidate the nature of the quencher, we have overlaid the SADS of $S_q$ with those associated with S1 and S* in Fig. 4a. The SADS of $S_q$ strongly resembles that of S* (black vs. red spectra in Fig. 4a), but not that of S1 (black vs. blue), in most of the region between 540 nm and 620 nm. Also, the lifetime of $S_q$ (12 ps, $84 ns^{-1}$) is two times slower than that of S1 (6 ps, $183 ns^{-1}$). An additional strong difference between the astaxanthin $S_q$ and S1 states lies in their role in EET within LHC–Asta: whereas the S1 state is found to be a donor of excitation energy to Chls (see above), the $S_q$ state is found to be an acceptor. Taken together, these findings indicate that $S_q$ corresponds to a Car excited state distinct from S1 and more similar to S*. The presence of multiple dark states (S1, S* and $S_q$), together with the fact that LHC–Asta contains only one Car species, indicate that astaxanthin is bound to the protein in different conformations. Some conformers then create a quenching site within LHC–Asta: the $S_q$ state. This state must be lower in energy than the Chls and/or possess a higher coupling with them. This last condition might derive from an increased dipole character of the $S_q \leftarrow S_0$ transition due to a higher degree of twisting of the Car[40]. The heterogeneity in the population rates of $S_q$ from the Chls (Fig. 3b, d) might also be related to the presence of multiple conformations of the astaxanthin bound to LHCs. A cartoon of the different dark states resolved in LHC–Asta is shown in Fig. 4b.

**Car conformational flexibility in LHCs.** We have shown that astaxanthin binds to LHC–Asta in at least three pools of conformations, of which one opens a quenching channel ($S_q$). For cyclic Cars, such as astaxanthin and the native xanthophylls bound to LHCs, rotation of the conjugated ring around the polyene chain can strongly modulate the properties of the first

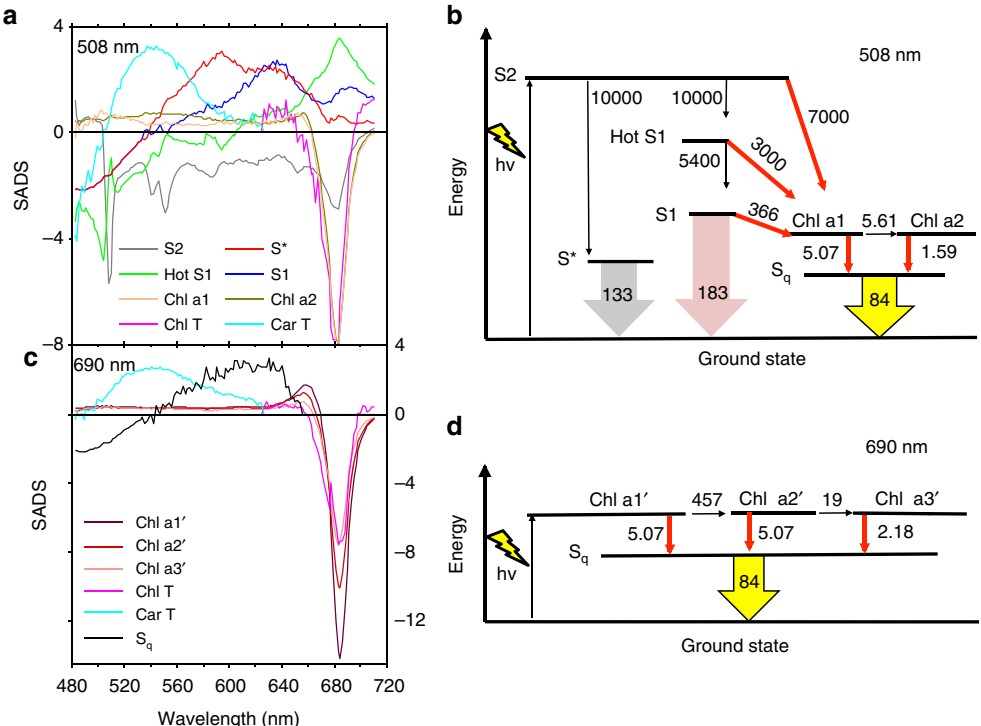

**Fig. 3** Target models of the excited state dynamics in LHC–Asta after Car (508 nm) and Chl (690 nm) excitations. **a** SADS estimated from the kinetic model for Car excitation (508 nm) shown in **b**. Spectrum and lifetime of $S_q$ have been constrained to be the same for both models and for clarity the associated SADS is not shown in **a**. **c** SADS estimated from the kinetic model for 690 nm excitation shown in **d**. In **b** and **d**, red arrows indicate the direction of EET between the various pigments. SADS of the Car and Chl triplet states (Chl T and Car T) are assumed to be zero above 630 nm and below 630 nm, respectively. The numbers next to the arrows in **b** and **d** are rates in ns⁻¹. For clarity, the components describing unconnected astaxanthin in **b**, and the formation of Chl T and Car T states are not shown in **b**. and **d**. The full target models are shown in Supplementary Figure 5

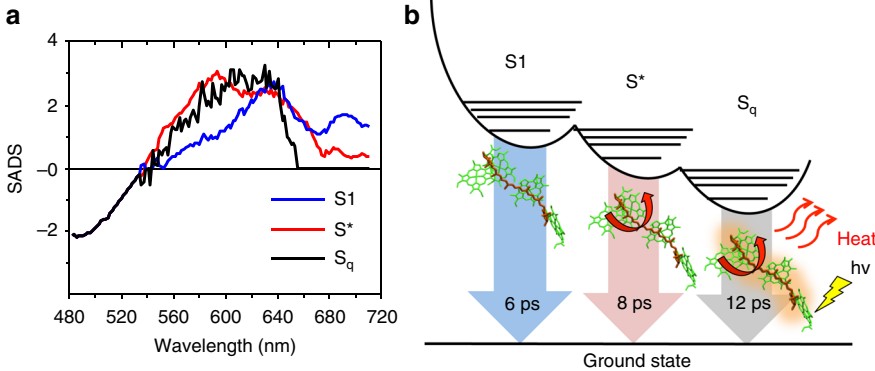

**Fig. 4** Dark states of astaxanthin bound to LHC–Asta. **a** Overlay of the SADS of the different Car dark states estimated from the target analysis reported in Fig. 3. **b** Simplified scheme of the potential energy surface of astaxanthin bound to LHC–Asta. In this representation, S1 is higher in energy than S* and $S_q$, in accordance with previously published information[41]. $S_q$ is positioned at the lowest level because it is found to be the only state able to quench Chls in LHC–Asta. For further explanation, see text. The red arrows indicate that $S_q$, similarly to S*[41, 68], originates from twisted conformations of the Car

excited state (spectrum, lifetime, and energy level)[41]. To investigate the conformational disorder of astaxanthin in LHC–Asta, we analyzed six independent molecular dynamics simulations of an astaxanthin-binding LHCII monomer embedded in a model lipid membrane (see Methods in the SI). Owing to the long sampling time (a total of ~5.5 µs), a large ensemble of LHCII conformations was observed. Over this ensemble, we quantified the conformational flexibility of astaxanthin bound to each of the two putative quenching sites, L1 and L2. To do so, we selected the angles between the end rings and the polyene chain and computed the distribution of these dihedral angles along the entire simulation trajectories.

We find that in both L1 and L2 sites on both sides of the membrane (stroma and lumen), astaxanthin shows a large conformational disorder (Supplementary Fig. 8) in agreement with our experimental observation of multiple dark states present in the complex and in particular with the presence of an S* state, which is predicted to originate from chain distortions[41]. Another possible interpretation for $S_q$, would be that this state is originated by an intramolecular charge transfer, as this state was shown to be active in other species of keto-Cars when these are hydrogen bonded to proteins[9,42,43]. To investigate this possibility for astaxanthin in LHC–Asta, we analyzed the probability for each keto group (six in total for astaxanthin in site L1, L2, and N1) to

form hydrogen bonds with the LHCII apoprotein. We found that only a single keto group formed a hydrogen bond and only a minor part of the overall simulated time (26% of the overall time in average, Supplementary Table 3). Such a low population of hydrogen bonded keto groups cannot explain the 70% overall quenching observed in LHC–Asta. Therefore, together with the experimental evidence that intramolecular charge transfer spectra are usually more red than the ones associated to S1[42–44], whereas $S_q$ and S* are found here to be ~40 nm blue-shifted, that the spectra associated to the dark state(s) of monomeric astaxanthin are independent on solvent[35] and that, such spectra, are almost identical to the ones observed here for LHC–Asta (Fig. 2a), we can exclude that $S_q$ originates from an intramolecular charge transfer state.

Next, we investigated the possibility to activate an $S_q$ state in wild-type LHCs. Indeed, it was shown spectroscopically that all native Cars bound to LHCs undergo a similar "twist" in solution[41]. It is unknown, however, whether such Car structural flexibility, which would enable a quenching mechanism similar to the one found in LHC–Asta, is favored within the Car-binding pocket of a LHC as well. To investigate the conformational disorder of native Cars when bound to LHCs, we analyzed six independent molecular dynamics simulations of a monomer of LHCII embedded in a model lipid membrane.

In Fig. 5, the distributions of the dihedrals associated to the two end rings of each lutein are shown together with the values computed over the three available LHCII crystals[16,27,45]. The values computed based on the crystal structures fall within the ranges sampled by the simulations and already show the high degree of disorder of the lutein molecules bound to LHCII. The largest distribution of angles is obtained at the lumen side, and in particular at the L2 site. This means that the probability to undergo a structural change in this domain is the highest. This is interesting considering that under strong light intensities, acidification of the lumen has been proposed to trigger (directly or indirectly) the conformational switch of LHCs[6,46,47]. Moreover, the ring of the lutein exposed to the lumenal side of the complex is the one conjugated to the polyene chain[16,27,45,48], therefore the rotation of this specific ring is responsible for the Car energetics. The fact that the site in which lutein experiences the major conformational freedom is exposed to the lumen and also contains the conjugated ring suggests that the orientation of lutein molecules within LHCs is functionally important for the regulation of light harvesting.

## Discussion

In this work, we have investigated the origin of the quenching processes taking place in LHC monomers from plants that bind Chls and a single Car, astaxanthin. LHC–Asta was found to be highly quenched. Due to the similarity in Chl organization between this complex and the wild-type LHCII, LHC–Asta is an interesting model system to study quenching in isolated LHCs in the absence of aggregation and denaturing conditions (detergent removal). Via femtosecond transient absorption spectroscopy combined with global and target analyses of the data acquired after preferential Car and Chl excitation, we disentangled the contribution of two different Car dark states (S1 and S*) from the overall spectral evolution of astaxanthin. The decays of these two states are well separated in time and energy, as previously reported for a variety of Cars[37,38,41]. In LHC–Asta, S1 is not responsible for the energy dissipation present in the complex, as proposed for other quenched LHC model systems[8,9]. Instead, it acts as a donor of excitation energy to Chls. We resolved a third Car dark state, here called $S_q$, which, based on the spectrum and the lifetime, is strongly reminiscent of S*. This state, differently than the other astaxanthin states, acts as an acceptor of excitation

energy from Chls and is responsible for the quenching present in this model LHC.

In sum, for the first time, different Car configurations are identified within the same LHC complex, and their role in light harvesting is shown to be distinct, suggesting that a structural change of these pigments may be involved in inducing photoprotection in plants. Based on MD simulations, we show that the structural switches that induce changes in the site energy and lifetime of their dark states are likely also present within wild-type LHCs. Notably, these switches are favored on the lumenal side of the complex where different light intensities trigger major changes in the physico-chemical environment. We, therefore, propose that structural changes in the LHC protein coupled to structural changes of the Cars represent an efficient strategy to cope with excess light energy in vivo, thus preventing photooxidative stress and damage to the photosynthetic apparatus.

## Methods

**Sample preparation.** Monomeric LHC fractions (LHC–Asta system) were isolated from astaxanthin-producing transplastomic tobacco (*N. tabacum*) plants[26] by sucrose density gradient centrifugation[31,49]. In brief, 0.2 mg total Chl of thylakoids were washed with 5 mM ethylenediaminetetraacetic acid and resuspended in 200 µl 10 mM N-(2-hydroxyethyl)piperazine-N'-ethanesulfonic acid (Hepes) at pH 7.5. An equal volume of 1.2% α-DDM (*N*-dodecyl-α-D-maltoside) was added, mixed gently, and the solubilized thylakoids were centrifuged at 14,000 rpm for 10 min at 4 °C. The supernatant was loaded on a 0–1 M sucrose gradient (10 mM Hepes, pH 7.5, 0.03% α-DDM) and centrifuged at 288,000 *g* for 17 h. The separated bands were collected with a syringe. The second band contained monomeric LHC–Asta.

For all measurements, the sample was diluted to the required OD in a buffer consisting of 10 mM HEPES (pH 7.6), 0.5 M sucrose and 0.03% α-DDM. Wild-type LHCII monomers (LHCII-WT) were reconstituted following the protocol previously reported[50]. In brief, recombinant His-tagged apoprotein of LHCII was expressed in *Escherichia coli* and purified as inclusion bodies. The denatured protein was reconstituted in vitro by adding pigments extracted from spinach leaf. Refolded LHCII was purified from the unbound pigments by Ni-affinity chromatography and sucrose gradient centrifugation.

Pigment content was analyzed by HPLC after 80% acetone extraction[51].

**Steady-state spectra and time-resolved fluorescence.** Room temperature absorption spectra were acquired on a Varian Cary 4000 UV-Vis spectrophotometer. Circular dichroism (CD) spectra were recorded on a Chirascan CD Spectrophotometer (Applied Photophysics), at 10 °C. Time-resolved fluorescence was recorded via Time-Correlated Single Photon Counting on a FluoTime 200 from PicoQuant, at 10 °C. Excitation was centered at 470 nm (Chl b and astaxanthin region), with an average power of ≈100 µW and a repetition rate of 10 MHz. Signal was accumulated until a maximum of 20 thousand counts at the peak channel, over channels separated by 8 ps. The Instrument Response Function was estimated via the measured decay of a pinacyanol iodide, whose lifetime is ≈6 ps[52].

**Femtosecond transient absorption.** Transient absorption spectroscopy experiments were conducted at room temperature on a system described elsewhere[53]. In brief, mode-locked 800 nm-seed pulses (Coherent-MIRA seed) were amplified via regenerative amplification (Coherent-Rega 9050) and split into pump and probe pathways (60%/40% ratio). The pump wavelength was tuned either to 508 or 690 nm via optical parametric amplification (Coherent OPA 9400), and the final pump pulse was narrowed around the central value with a FWHM of 10 nm by means of interference filters (THORLABS). The probe pulse was focused into a sapphire crystal to generate white-light continuum. The dispersed probe pulse is detected via a 76-channels photodiode array. Each spectral window covers a range of ≈130 nm. For each excitation and each power, two spectral windows were recorded, in the Soret region (window range ≈480–610 nm) and the Chl–$Q_X$/$Q_Y$ region (window range ≈590–720 nm), respectively. A delay line varies the pump-pathway length, allowing to record transient absorption spectra until 3.5 ns after excitation. All experiments were run at a repetition rate of 40 kHz. OD values of the sample were set to ≈0.6 cm$^{-1}$ at the peak around 680 nm, in a 1 mm quartz cuvette. A shaker ensured refreshing of the sample throughout each measurement this way preventing sample degradation during the ultrafast experiments, as verified by the absence of changes in sample absorption.

Selective excitation of astaxanthin (508 nm) was collected under a 5 nJ per pulse pump energy. Regarding the Chl excitation (690 nm), the data presented in the main manuscript for the global analysis are mainly from experiments at 15 nJ per pulse pump energy where singlet-singlet annihilation effects[54] are largely prevented given the low absorption cross section of LHC–Asta at 690 nm (Supplementary Fig. 1) and the power used (a flux of ≈10$^{14}$ photons per cm$^2$ corresponding to less than one absorbed photon per complex). To further quantify the eventual presence

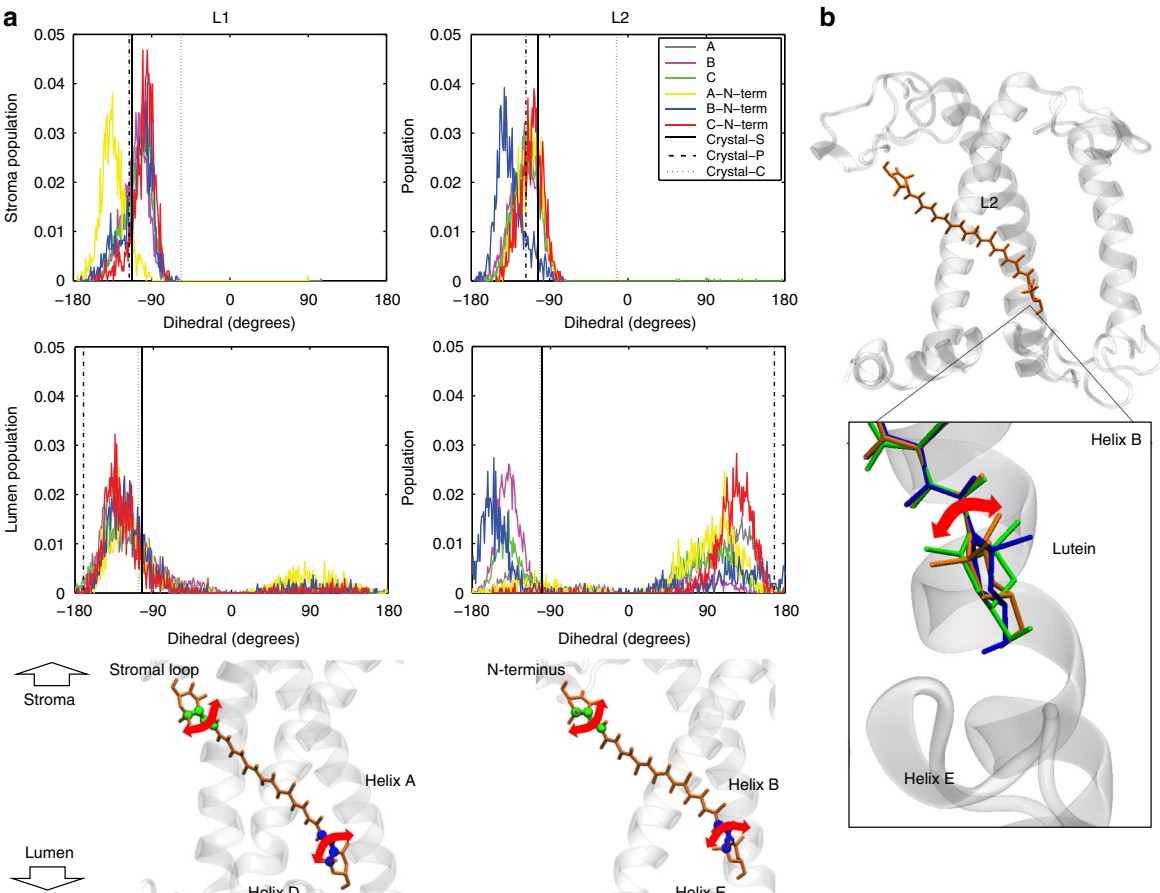

**Fig. 5** Conformational flexibility of the Cars' end rings in wild-type LHCII. **a** The dihedral distributions for the two end rings of lutein bound to site L1 and L2 of LHCII were computed over six independent MD simulations, discarding the first 400 ns of simulation per each run. The different simulations are labeled as in our previous work[7]. The values for the dihedrals measured on three different crystals of LHCII are also reported and are indicated in the plots as Crystal-S, P or C according to the organism from which LHCII was purified (spinach[27], pea[45] or cucumber[16]). The dihedrals of the ring located at the stromal side are reported on the top panels, and those of the ring at the lumenal side are presented in the middle panels. The color scheme for all the plots follows the legend inserted in the plot associated to the L2 site, stromal side (upper right panel of **a**). The lower panels show the atoms used to define this dihedral angle (green for the stromal and blue for the lumenal side of the membrane). Cars are shown as orange sticks. The nearby elements of LHCII apoprotein are shown in transparent white. The left panels are for site L1 and the right for site L2. **b** A representation of LHCII apoprotein and of the lutein in the L2 site (depicted as in **a**) and, in the inset, an example of the rotations of the end-ring of lutein observed at the L2 site in one of the simulations. Different lutein conformations are depicted in different colors and aligned on top of each other

of singlet-singlet annihilation effects, we also collected data at lower energies (5, 10 nJ), as described in the caption of Supplementary Table 2. As reported in the same table, the contribution of annihilation to the kinetics is very minor, ranging between 1% and 14% for the powers used for these experiments (5–15 nJ).

**MD simulations**. In this work, we present the results of 12 independent classic MD simulations of a monomer of LHCII embedded in a model lipid membrane with explicit solvent in the microsecond timescale and with atomistic resolution. All MDs were run on the GROMACS simulation package, version 4.6.3 (http://www.GROMACS.org/55). Simulations were analyzed using the tools available on the GROMACS platform and on the VMD software (http://www.ks.uiuc.edu/Research/vmd/[55]).

As in Ref. [7], all the simulations are based on the crystal structure of a monomeric subunit of the LHCII (PDB 1RWT, Chain A)[27]. The monomeric protein (232 amino acids, residues 14–246) has been simulated altogether with its cofactors: eight chlorophyll a (Chl a), six chlorophyll b (Chl b), and two Luteins (Lut 1 and 2), 1 Violaxanthin (Vio), 1 Neoxanthin (Neo), in the case of the wild-type, or three astaxanthin molecules in the case of LHC–Asta. In addition, present are 60 interstitial water molecules (at the start of the simulation) and one DPPG (1,2-dipalmitoyl-sn-glycero-3-phosphoglycerol) lipid molecule. The simulations here described for the LHCII wild-type monomer are the same as in ref. [7], whereas the ones of LHC–Asta represent a new set of simulations.

The force field used for all the simulations is a united-atom force field (GROMOS 54a7[56]), where all the titratable amino acids were considered to be in the standard protonation state at pH 7. The force–field parameters for the Chls, Cars, and DPPG have been derived compatibly with the GROMOS 53a6 force field, as previously reported[7,57]. The force–field parameters for astaxanthin are identical

to the ones of lutein but for two differences: the presence of two conjugated rings (astaxanthin) instead of one (lutein), and on the presence of two keto groups, each on one ring (astaxanthin), which are missing in lutein. Parameters and charge used for the keto groups are the ones already parametrized for the keto groups on the ring of Chls[57]. Parameters compatible to two conjugated rings have been taken into account for astaxanthin.

The LHCII wild-type was embedded into a pre-equilibrated 1-palmitoyl-2-oleoyl-sn-glycero-3-phosphocholine (POPC) bilayer via the multi-step protocol previously described[7]. In brief, a coarse grained (MARTINI[58]) system consisting of LHCII apoprotein, POPC membrane (two homogeneous layers of pure POPC in the ratio 204:204 per monolayer) and 5376 MARTINI water beads were minimized (500 steps of steepest descent) and then relaxed (30 ns, NPT) at 323 K. From the last step of the trajectories we retrieved the membrane coordinates and converted the structure to united-atom resolution (GROMOS 54a7[56]) via the SUGARPIE tool[59]. The GROMOS-LHCII pigment–protein complex structure (wild-type) was then aligned to the pore present in the GROMOS-POPC bilayer (via editconf tool of GROMACS[60], resulting in a final bilayer of 344 POPC molecules) and consequently solvated the system with water (>15 k molecules) and ions (neutral physiologic conditions, 10 mM Na⁺ Cl⁻[61]).

The system was first minimized (steepest descent) and relaxed (10 ps, NVT + 40 ns NPT). During all these steps, isotropic strong position restraints were applied to the protein (backbone) and to all its ligands (Chl–tetrapyrroles and whole Cars and DPPG molecule structures) to avoid perturbation of the protein and cofactors crystal positions. Position restraints were set to a starting value of 10,000 kJ mol⁻¹ nm⁻² and then gradually reduced to zero every 10 ns of NPT simulation (for a total of 40 ns simulated time), following the sequence $1000 \rightarrow 1000 \rightarrow 500 \rightarrow 200$ kJ mol⁻¹ nm⁻².

From the final snapshot retrieved after the last 10 ns at 200 kJ mol$^{-1}$ nm$^{-2}$, we started 12 independent simulations (A, B, C, A-Nterm, B-Nterm, C-Nterm for the wild-type, A to F for the LHC–Asta). Concerning the wild-type, as described in ref. [7], from this final snapshot we started either three independent simulations from different initial random velocities (A to C) or three independent control simulations (A-, B-, C- Nterm) where the N terminus (first 39 protein residues) was allowed to relax for additional 100 ns (all cofactors and the protein backbone were kept constrained with a force constant of 200 kJ mol$^{-1}$ nm$^{-2}$). Concerning LHC–Asta, starting from the coordinates of the above mentioned snapshot (retrieved after the last 10 ns at 200 kJ mol$^{-1}$ nm$^{-2}$), via the software CHIMERA[62] we added the coordinates of the keto groups to the two central luteins (site L1 and L2) and shifted the position of one double bond on the stromal ring. Regarding site N1, after aligning the backbone of LHCII from this snapshot with the one of Lhca-1 from photosystem I[63], we used the coordinates of the all-trans betacarotene in site N1 of Lhca-1 to re-build the structure of astaxanthin, again via the CHIMERA software[62]. From this structure, additional 10 ns with position restraints of 200 kJ mol$^{-1}$ nm$^{-2}$ on protein backbone, Chl–tetrapyrroles and whole astaxanthin and DPPG molecule structures were run to allow equilibration of the new structures. After this, six independent simulations (A to F) were run each for over 800 ns, for a total of ~ 5.5 µs.

For all simulations, we used an integration time step of 2 fs with constraints on all the bonds (LINCS algorithm[64]). For long-range electrostatics, the Particle Mesh Ewald scheme was used with cutoff values of 1 nm and 1.4 nm, respectively, for short-range Coulomb and van der Waals interactions. Via a Parrinello-Rahman barostat[65], constant pressure was set to 1 bar under semi-isotropic coupling with relaxation time constant of 5 ps and compressibility of $4.5 \times 10^{-5}$ bar$^{-1}$. Via a Nose-Hoover thermostat-scheme[66], constant temperature was set to 300 K with 0.5 ps time constant and with solvent, membrane and LHCII complex coupled to the thermostat. For all the simulations, Periodic Boundary Conditions were employed.

**Data availability**. The authors declare that all data supporting the findings of this study are included in the main manuscript file or Supplementary Information or are available from the corresponding author upon request.

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

## Acknowledgements

We would like to thank John Kennis, Joris Snellenburg and Vincenzo Mascoli for insightful discussions on the data shown in the manuscript. We are grateful to Alex de Vries for his expert advices on deriving the force–field parameters for astaxanthin and to Zhenfeng Liu and Mei Li for providing the PDB file of the crystal of LHCII from cucumber. This work is supported by The Netherlands Organization for Scientific Research (NWO), division Earth and Life science (ALW) via a Vici grant and division Foundation for Fundamental Research on Matter (FOM) and by the ERC consolidator grant 281341 (ASAP) to R.C., and grants from the European Union (EU-FP7 DISCO 613513) and the European Research Council (ERC) under the European Union's Horizon 2020 research and innovation programme (ERC-ADG-2014; grant agreement No 669982) to R.B. This work was carried out on the Dutch national e-infrastructure with the support of SURF Cooperative through a NWO Pilot-grant to N.L. and R.C.

## Author contributions

N.L. and R.C. conceived and designed the research. Y.L., D.K. and R.B. produced the transplastomic tobacco. P.X. purified the samples, performed biochemical analysis, measured the steady-state spectra, and performed time-resolved fluorescence measurements. N.L. performed transient absorption experiments, preliminary global and target models and in silico analysis. N.L., I.H.M.vS., B.vO. and R.C. designed the complete target model and I.H.M.v.S. performed the final version of the target analysis. N.L. prepared all the figures and wrote the manuscript with contributions from I.H.M.vS, B. vO, and R.B. and R.C. All authors approved the final version of the manuscript.
