## [Peer Review File · Nature Communications]

Reviewers' comments:

Reviewer #1 (Remarks to the Author):

This paper investigates Fl quenching in an LHC2 complex that contains a novel carotenoid. It is in two parts a set of time resolved expts. and a simulation. I have a series of questions that relate to several major assumptions that are not clearly justified. A set of Car absorption changes that have slightly different difference spectra and clearly different kinetics are seen. it is not clear how these can be unequivocally assigned to different dark electronic states. This is a perennial problem with carotenoids. Then there is the idea that these have different energy levels. Again their seems to be no justification for this. Finally it is a big jump to go from molecular mechanic simulations to suggest different car configurations to say these will give rise to different electronic dark states with different energy levels. There is no evidence for this hypothesis either here or in the literature. These fundamental questions need to be dealt with to make this paper a strong and useful contribution.

Reviewer #2 (Remarks to the Author):

This article presents ultrafast spectroscopic experiments showing decay times associated with wild-type and astaxanthin-containing LHCII. By comparing these, they deduce different excitation energy transfer (chlorophyll->carotenoid) time scales in the two cases. These are used to construct a complex kinetic model and as a basis for speculation about the molecular mechanisms involved. This is surely interesting work which will appeal to a broad audience of both experimentalists and theoreticians. There is a danger that the complex kinetic model is overinterpreted, but the authors make this clear by proposing multiple consistent (e.g. serial and parallel) models. The molecular dynamics models are not very well integrated in the manuscript, but simply serve to indicate that there is conformational flexibility regarding the ring on one side of the carotenoid. This flexibility is then implicated as the reason for decay of EET. This is quite possibly correct and would not be surprising. But the connection between the conformational flexibility in the MD and signal decay in the experiment is not made clearly (and would require much more complicated simulations that computed energy transfer rates). I think this is fine for the present article, although the authors might be a little more upfront about this. The biggest problem with this article is that the MD simulations are not really described at all. We are referred to Ref. 7, but as far as I can tell, the simulations in the present paper are distinct from those in Ref. 7. There should be a brief summary of what simulations were carried out, what the boundary conditions were, what the length of the simulations was, etc. While it might be helpful to refer to Ref. 7 for details and rationale about the simulations, the methods section of this paper should be self-contained (at least for an expert in the MD field). Right now, even an expert cannot tell what was done (even after reading Ref. 7 which should not be required). This article needs to have the above issues attended to and then to be reviewed again. I believe it will ultimately be a nice contribution to Nature Comm, but this cannot be decided until the MD methods is completed.

Reviewer #3 (Remarks to the Author):

This is a timely manuscript describing the results of femtosecond transient absorption study of isolated LHCII complexes that contained one type of carotenoid but with different excited state lifetimes. The results of the study are interpreted in terms of existence of various conformational states in the complex that promote either light harvesting or photoprotective mode of antenna - fundamental for the fine tuning of the photosystem II efficiency. The work is sound and done on a very high instrumental level and I do not see any problems that would prevent this

work from publishing.

Reviewer #4 (Remarks to the Author):

A very interesting manuscript shedding some light on regulation of energy flow through the photosynthetic antenna. The authors have explored quite unique LHC containing the carotenoid astaxanthin that is not naturally present in LHCs of plants, algae or cyanobacteria. The major advantage is that this LHC contains a single-carotenoid species, situation that do not normally occur in LHCs from plants, allowing to study the role of conformational changes in energy transfer through the complex. The basic spectroscopic characterization by absorption and CD spectroscopy proves that LHC retains its natural structure and pigment organization, making it an ideal model to explore energy transfer pathways. The transient absorption data and fitting are quite solid, yet there are some issues that should be clarified.

1. On p. 4 the authors state that three carotenoid binding sites in LHC -Asta are occupied by the carotenoid. This however means that about one third of astaxanthin molecules should be in 9'-cis configuration, because only 9'-cis isomers can bind to the N1 site. I assume that the authors run HPLC to determine pigment composition thus they should be able to separate all-trans and 9'-cis astaxanthin in HPLC. Does the HPLC confirm the presence of 9'-cis astaxanthin?

2. In relation to the point above, I do not understand the statement in line 101, p. 4 saying that the difference in Chl-b absorption peak in abs. spectrum of LHC-Asta is due to absence of neoxanthin (that is in the site N1 in WT LHCI). Since the authors assume that astaxanthin binds to the N1 site, the Chl-b should be affected in a similar way as it is by the presence of neoxanthin. Or is there some really specific (covalent?) interaction between neoxanthin and Chl-b that cannot be achieved by astaxanthin?

3. I do not agree with the assignment of the 8 ps component in EADS (and the red SADS in Fig. 3a) to an S* state. Somehow, it looks like if there is no obvious assignment, the mysterious S* will help...First, the S* identified in long carotenoids never contains two bands, one of which coincides with the S1-Sn peak. This is precisely the case here and that should already indicate that it is likely not the S* state. Second, the high-energy peak in the "S* spectrum in Fig. 3a" peaks around 600 nm which is far too red to be assigned to the S* state. In fact, in the Supporting information in Ref. 33 it is shown that the S* state of astaxanthin in DMSO peaks at 565 nm, thus quite off the peak assigned to the S* state here.

The two peaks in transient absorption spectra, at 600 and 640 nm rather resemble the spectral response of hydroxyechinenone in the orange carotenoid protein (Polivka et al. Biochemistry 2005). There, binding of hydroxyechinenone also induces a transient absorption spectrum with two peaks even though in solution the carotenoid has only one S1-Sn band in transient absorption spectrum. The authors completely ignore the fact that astaxanthin is a keto-carotenoid thus interaction with the protein binding site can activate an intramolecular charge transfer (ICT) state that is characteristic of these carotenoids. Thus, I would rather assign the red and blue SADS in Fig. 3a to two S1/ICT states of astaxanthin having different degree of charge transfer character. The presence of the conjugated keto groups of astaxanthin is somehow a disadvantage here as it may induce effects that can never occur in WT LHCI.

4) Why the MD simulations were not carried out with astaxanthin? I assume it should be possible to take the LHCI structure and place astaxanthin molecules into the L1, L2 and N1 sites to see whether the effect of ring rotation occurs also for astaxanthin. Further, it could tell something about the possible interaction of the keto-oxygens with protein and maybe even provide some answers to my

question about the possible involvement of the ICT state. Could this be added?

Minor issues:

a) I assume that in Fig. 1d the y-axis is in log scale. It should be indicated in the figure.

b) p. 7, line 163: hot S1 172 fs while 170 fs is in Fig. 2a

c) Unclear why in Fig. 2 EADS are used to characterize data after 508 nm excitation while SADS are used for 690 nm excitation. This is quite confusing for a reader because these two figures are not visually comparable. Use either EADS or SADS, but use the same in Fig. 2a and 2b

d) Transfer times and rates are used in text and figures. The authors should decide whether they want to use rates or transfer times, but they should be consistent throughout the manuscript. It is confusing to read about transfer times in ps in the text, but to see only rates in Fig. 3.

Reviewer #1 (Remarks to the Author):

This paper investigates Fl quenching in an LHC2 complex that contains a novel carotenoid. It is in two parts a set of time resolved expts. and a simulation. I have a series of questions that relate to several major assumptions that are not clearly justified. A set of Car absorption changes that have slightly different difference spectra and clearly different kinetics are seen. it is not clear how these can be unequivocally assigned to different dark electronic states. This is a perennial problem with carotenoids. Then there is the idea that these have different energy levels. Again their seems to be no justification for this.

We experimentally show that there are two carotenoids dark states with different functions and different spectra (their maxima are separated by 43 nm). One of them acts as an energy donor for the chlorophylls and the other as a Chl quencher.

As described in the manuscript, the assignment is done by comparing the spectra and lifetimes of these states with literature data. For example the spectrum of the S1 state obtained here is virtually identical to that of the S1 state of Asta in DMSO (Fuciman, M. et al Chem Phys lett 2013), and the state attributed to S* has also the properties (spectrum and lifetime) expected for S* (Papagiannakis, E. et al PNAS 2002, Gradinaru, C et al PNAS 2001). The bases for the assignment are described in the manuscript.

Moerver, the essential point is that the data show that the two dark states have different functions, one is active as energy donor for the chlorophylls and the other as energy acceptor.

Regarding the energy levels, we explicetely explain how the different functions are caused by either a different energy level of the two states or a different level of coupling with the Chls, due to twisting of the Car introducing a higher dipole character in the transition, as already discussed in previous literature (e.g. Papagiannakis, E. et al PNAS 2002, Kloz, M. et al PCCP 2016).

Finally it is a big jump to go from molecular mechanic simulations to suggest different car configurations to say these will give rise to different electronic dark states with different energy levels. There is no evidence for this hypothesis either here or in the literature. These fundamental questions need to be dealt with to make this paper a strong and useful contribution.

The evidence for the existence of different Cars dark states and for their link to specific Car structures, as explained in the text, is supported by many theoretical and experimental works from several groups (e.g. Niedzwiedzki, D. M. et al JPCB 2006, Tavan, P. and Schulten, K J Chem Phys 1986, Billsten, H.H. et al Chem Phys Lett 2002, Gradinaru et al PNAS 2001, Papagiannakis, E. et al PNAS 2002, Kloz, M. et al PCCP 2016, Ostroumov, E. E. Science 2013, Hauer, J. JPCA 2013).

Reviewer #2 (Remarks to the Author):

This article presents ultrafast spectroscopic experiments showing decay times associated with wild-type and astaxanthin-containing LHCI. By comparing these,

they deduce different excitation energy transfer (chlorophyll->carotenoid) time scales in the two cases. These are used to construct a complex kinetic model and as a basis for speculation about the molecular mechanisms involved. This is surely interesting work which will appeal to a broad audience of both experimentalists and theoreticians.

There is a danger that the complex kinetic model is overinterpreted, but the authors make this clear by proposing multiple consistent (e.g. serial and parallel) models. The molecular dynamics models are not very well integrated in the manuscript, but simply serve to indicate that there is conformational flexibility regarding the ring on one side of the carotenoid. This flexibility is then implicated as the reason for decay of EET. This is quite possibly correct and would not be surprising. But the connection between the conformational flexibility in the MD and signal decay in the experiment is not made clearly (and would require much more complicated simulations that computed energy transfer rates). I think this is fine for the present article, although the authors might be a little more upfront about this. The biggest problem with this article is that the MD simulations are not really described at all. We are referred to Ref. 7, but as far as I can tell, the simulations in the present paper are distinct from those in Ref. 7. There should be a brief summary of what simulations were carried out, what the boundary conditions were, what the length of the simulations was, etc. While it might be helpful to refer to Ref. 7 for details and rationale about the simulations, the methods section of this paper should be self-contained (at least for an expert in the MD field). Right now, even an expert cannot tell what was done (even after reading Ref. 7 which should not be required). This article needs to have the above issues attended to and then to be reviewed again. I believe it will ultimately be a nice contribution to Nature Comm, but this cannot be decided until the MD methods is completed.

We have now added 6 simulations of LHCII containing astaxanthin. These simulations show the high flexibility of Asta in the binding sites and provide the link between the experimental work on LHCII-Asta and the MD on LHCII-WT. The methods are now added to the ms as suggested by the reviewer.

Reviewer #3 (Remarks to the Author):

This is a timely manuscript describing the results of femtosecond transient absorption study of isolated LHCII complexes that contained one type of carotenoid but with different excited state lifetimes. The results of the study are interpreted in terms of existence of various conformational states in the complex that promote either light harvesting or photoprotective mode of antenna - fundamental for the fine tuning of the photosystem II efficiency. The work is sound and done on a very high instrumental level and I do not see any problems that would prevent this work from publishing.

We are glad the reviewer appreciates our work.

Reviewer #4 (Remarks to the Author):

A very interesting manuscript shedding some light on regulation of energy flow through the photosynthetic antenna. The authors have explored quite unique LHC containing the carotenoid astaxanthin that is not naturally present in LHCs of plants, algae or cyanobacteria. The major advantage is that this LHC contains a single-carotenoid species, situation that do not normally occur in LHCs from plants, allowing to study the role of conformational changes in energy transfer through the complex. The basic spectroscopic characterization by absorption and CD spectroscopy proves that LHC retains its natural structure and pigment organization, making it an ideal model to explore energy transfer pathways. The transient absorption data and fitting are quite solid, yet there are some issues that should be clarified.

1. On p. 4 the authors state that three carotenoid binding sites in LHC-Asta are occupied by the carotenoid. This however means that about one third of astaxanthin molecules should be in 9'-cis configuration, because only 9'-cis isomers can bind to the N1 site. I assume that the authors run HPLC to determine pigment composition thus they should be able to separate all-trans and 9'-cis astaxanthin in HPLC. Does the HPLC confirm the presence of 9'-cis astaxanthin?

The binding in the N1 site does not require cis-carotenoids. All-trans β -carotene is present in the N1 site of the Lhca complexes (Qin et al. Science 2015; Mazor et al. Elife 2015). In addition, the MD simulations show that all-trans astaxanthin is stably bound to the N1 site of LHCII (Table S3, added to revised ms).

2. In relation to the point above, I do not understand the statement in line 101, p. 4 saying that the difference in Chl-b absorption peak in abs. spectrum of LHC-Asta is due to absence of neoxanthin (that is in the site N1 in WT LHCII). Since the authors assume that astaxanthin binds to the N1 site, the Chl-b should be affected in a similar way as it is by the presence of neoxanthin. Or is there some really specific (covalent?) interaction between neoxanthin and Chl-b that cannot be achieved by astaxanthin?

The changes in the Chl b spectrum are minor; we made this more clear in the revised manuscript. It is likely that the interactions between the neo and the surrounding Chls are different from that of Asta and the same Chls and this can explain the difference in the spectra. However the change in absorption is negligible and the overall organization of the Chls, and in particular of the lowest energy forms which are Chl a, is not affected as confirmed also by the CD spectrum.

3. I do not agree with the assignment of the 8 ps component in EADS (and the red SADS in Fig. 3a) to an S^* state. Somehow, it looks like if there is no obvious assignment, the mysterious S^* will help...First, the S^* identified in long carotenoids never contains two bands, one of which coincides with the S1-Sn peak.

This is precisely the case here and that should already indicate that it is likely no the S^* state. Second, the high-energy peak in the " S^* spectrum in Fig. 3a" peaks

around 600 nm which is far too red to be assigned to the S* state. In fact, in the Supporting information in Ref. 33 it is shown that the S* state of astaxanthin in DMSO peaks at 565 nm, thus quite off the peak assigned to the S* state here.

The “single-band spectrum” is not strictly the rule for S*. For example, in Gradinaru, C. et al PNAS 2001 and Niedzwiedzki, D. M. et al JPCB 2006 S* clearly shows multiple peaks.

The spectra obtained from global analysis, EADS, in our manuscript and in Asta aggregates (Fig. 2.a in our ms and Fig. 3.a in ref. 33: Fuciman et al. Chem Phys lett 2013) are nearly identical in the carotenoid region and represent a mix of states. Our target analysis enabled us to resolve all the entangled components (S1 and S* for example). In ref 33 such analysis is absent, making it impossible to compare the spectrum of our S* pure species to the ones convoluted under the EADS presented in ref 33. The species that we assign to S* has also a lifetime (12 ps) more consistent with all experimental findings on S* (e.g. Niedzwiedzki, D. M. et al JPCB 2006, Gradinaru et al PNAS 2001, Papagiannakis, E. et al PNAS 2002, Kloz, M. et al PCCP 2016), than the one assigned to S* in ref 33 (33 ps).

The two peaks in transient absorption spectra, at 600 and 640 nm rather resemble the spectral response of hydroxyechinenone in the orange carotenoid protein (Polivka et al. Biochemistry 2005). There, binding of hydroxyechinenone also induces a transient absorption spectrum with two peaks even though in solution the carotenoid has only one S1-Sn band in transient absorption spectrum. The authors completely ignore the fact that astaxanthin is a keto-carotenoid thus interaction with the protein binding site can activate an intramolecular charge transfer (ICT) state that is characteristic of these carotenoids. Thus, I would rather assign the red and blue SADS in Fig. 3a to two S1/ICT states of astaxanthin having different degree of charge transfer character. The presence of the conjugated keto groups of astaxanthin is somehow a disadvantage here as it may induce effects that can never occur in WT LHCII.

Several evidences indicate that S_q/S^* is not an ICT: first, ICT states in all carotenoids are more red than S1, while S_q/S^* are blue-shifted by over 40 nm, as widely shown (e.g. Polivka, T. et al BBA 2013, Slouf, V. et al PNAS 2012, Zigmantas, et al PCCP 2004). Second, as specified above, our EADS after Asta-excitation are identical to the ones measured by Fuciman et al. (Chem Phys lett 2013), and in this same paper the authors observe that excited state dynamics are independent on solvent and therefore conclude that the presence of an ICT in astaxanthin can be excluded.

Additionally, as suggested by the reviewer (see below), we have run MD simulations for LHCII-Asta and we have verified that only 1 over the 6 keto-groups present in LHC-Asta forms an hydrogen bond with the protein, and for only 26% of the time, which is insufficient to explain a 70% overall quenching in our complex. This is now added to the revised manuscript.

4) Why the MD simulations were not carried out with astaxanthin? I assume it should be possible to take the LHCII structure and place astaxanthin molecules into the L1, L2 and N1 sites to see whether the effect of ring rotation occurs also

for astaxanthin. Further, it could tell something about the possible interaction of the keto-oxygens with protein and maybe even provide some answers to my question about the possible involvement of the ICT state. Could this be added?

As suggested by the reviewer we added an extensive set of MD simulations of LHC-Asta for a total of ~5.5 microseconds. We do find a high probability of ring rotations in all the putative quenching sites confirming our experimental observations.

Minor issues:

a) I assume that in Fig. 1d the y-axis is in log scale. It should be indicated in the figure.

Corrected

b) p. 7, line 163: hot S1 172 fs while 170 fs is in Fig. 2a

Corrected

c) Unclear why in Fig. 2 EADS are used to characterize data after 508 nm excitation while SADS are used for 690 nm excitation. This is quite confusing for a reader because these two figures are not visually comparable. Use either EADS or SADS, but use the same in Fig. 2a and 2b

In Fig. 2a-b we show EADS (a) and DADS (b) because this is the most clear way to identify the similarities in the transient spectrum of the quencher (b) with the one of singlet excited states of astaxanthin (a). We still think that this is a more clear comparison and in the caption we explain that the two figures describe different type of spectra.

d) Transfer times and rates are used in text and figures. The authors should decide whether they want to use rates or transfer times, but they should be consistent throughout the manuscript. It is confusing to read about transfer times in ps in the text, but to see only rates in Fig. 3.

Wherever we mention the transfer times, we now added the rates as well.

REVIEWERS' COMMENTS:

Reviewer #2 (Remarks to the Author):

The authors have addressed my concerns and included simulations of LHCII-Asta

Reviewer #4 (Remarks to the Author):

The authors have reasonably responded to all my comments and questions. I appreciate the additional MD simulations with astaxanthin as it provides certainly a clearer picture what could be going on in this unique system. It is actually interesting that astaxanthin exhibits even larger conformational flexibility than lutein. I believe that the revised manuscript meets the high quality standard required for publications in Nat. Comm.